# Applying the Pender’s Health Promotion Model to Identify the Factors Related to Older Adults’ Participation in Community-Based Health Promotion Activities

**DOI:** 10.3390/ijerph18199985

**Published:** 2021-09-23

**Authors:** Hsuan-Hui Chen, Pei-Lin Hsieh

**Affiliations:** 1Nursing Department, Yang Ming Hospital, Taoyuan City 324005, Taiwan; jimrainc@yahoo.com.tw; 2School of Nursing, Chang Gung University of Science and Technology, Chang Gung Medical Foundation, Associate Research Fellow, Taoyuan City 33303, Taiwan

**Keywords:** Pender’s health promotion model, older adults, community-based health promotion activities, health promotion questionnaire

## Abstract

Aging societies have garnered global attention regarding issues related to older adults’ health promotion. This cross-sectional study aimed to identify factors associated with older adults’ participation in community-based health promotion activities. The Older Adults’ Health Promotion Activity Questionnaire was developed to collect data, and a total of 139 older adults were recruited from a community care center in Taoyuan City. Participants’ mean age was 72.7 years (SD = 6.0 years), 74.8% were females, 64.7% were married, 59% had a lower level of education, 51.8% had lower income, 59% reported their health status not good, and 76.3% had chronic disease. Our findings indicated that age, perceived benefits, and self-efficacy were identified as significant predictors of participation in health promotion activities. Among them, perceived benefits were found to have the strongest association with participation in health promotion activities (β = 0.305; *p* < 0.05). The findings showed that the Pender’s Health Promotion Model is useful to provide information for predicting and detecting significant factors related to older adults’ participation in community-based health promotion activities. By using this model as a framework, researchers can design more specific studies that are directed towards improving healthy lifestyles and detecting the key components of health-related behaviors among different age groups.

## 1. Introduction

The proportion of older adults (aged 65 and over) is gradually increasing in countries around the world because of declining fertility rates. It is expected that the global population of older adults will reach 2 billion by 2050 [1]. By the end of 2019, the aging index of Taiwan, which has been on the rise, reached 119.8; this value is higher than the benchmark value for an aging society according to the World Health Organization (WHO). More than 30% of older adults in Taiwan have two or more chronic conditions, such as diabetes, heart failure, arthritis, or dementia, that can lead to increased hospitalizations or nursing home stays [2]. Managing one’s health and being involved in health-promoting activities can have a positive impact on health, mortality, and quality of life.

Community-based health promotion activities have multiple benefits to improve older adults’ physical functions, spiritual satisfaction, and sense of accomplishment [3]. Engagement in health promotion activities lasting 12 weeks or longer is a critical component to lasting health effects as such activities have been reported to have significant positive effects on older adults’ physical, mental, and spiritual well-being [4,5,6]. Participation in health promotion activities is influenced by one’s cognition, experience, family, society, culture, etc. [7]. Important factors affecting older adults’ involvement in community-based health promotion activities include personal characteristics, perceived activity benefits, perceived activity barriers, perceived activity self-efficacy, situational factors, interpersonal relationship factors, and feelings toward activity participation [4,6,8,9]. Identifying the effective factors on participation in health promotion activities in older adults is necessary to improve this behavior [3].

The Health Promotion Model identifies factors that influence health behaviors. Pender’s Health Promotion Model is one of the most widely used models to identify and change unhealthy behaviors and promote health [10,11]. Predicting factors and explanatory constructs of health behavior in Pender’s model include perceived benefits, barriers, and self-efficacy; behavioral emotions; and interpersonal and situational influencers [12]. The various constructs have been introduced as the strongest predictors of nutritional and self-care behaviors in recent studies [13,14]. The reason for emphasizing the use of Pender’s Health Promotion Model is because this model explores, from a theoretical perspective, the factors and relationships that contribute to participation in community-based health promotion activities and enhanced health and quality of life among older adults.

Previous studies have suggested that older adults’ participation in health promotion activities is closely related to perceived activity benefits and barriers. In terms of perceived benefits, older adults have reported that participating in health promotion activities enhanced their overall mind–body fitness and physical conditions [7,8,15]. In addition, social support when participating in health promotion activities not only positively affects the physical and mental health of older adults but also plays an important role in reinforcing their continuous involvement in such activities [7,8]. In contrast, various activity barriers can adversely affect their enthusiasm in health promotion activities. Perceived activity barriers are negatively correlated with older adults’ involvement in health promotion activities [4], while those with fewer perceived activity barriers are more likely to participate in health promotion activities [16]. In Pender’s Health Promotion Model, the concept of perceived self-efficacy is included as part of behavior-specific cognitive and emotional factors. This is supported by multiple domestic and international studies, which reported that self-efficacy is an important factor that promotes an individual’s participation in health promotion behaviors and lifestyle [10,17,18,19].

Pender’s Health Promotion Model has been widely adopted to explore different health promotion behaviors [10,11,19] and has achieved concrete results. However, limited studies have utilized this model to investigate older adults’ engagement in community-based health promotion activities. Therefore, this study aimed to apply this model to identify the factors associated with older adults’ participation in community-based health promotion activities.

## 2. Materials and Methods

### 2.1. Study Design

A cross-sectional design was employed, and a questionnaire survey was conducted at a community care center in Taoyuan City, Taiwan. The community care center operates during daytime hours (9am–4pm), Monday through Friday. The center provides health promotion activities including stretching or other gentle exercise, mental stimulation games such as bingo, creative expression through arts and crafts, and nutritious meals. At present, there are about 250 older adults who use this center. Data were collected between January and April 2020.

### 2.2. Participants

A total of 139 older adults were recruited from a community care center in Taoyuan City. The inclusion criteria were participating in health promotion activities once a week and continuous participation for more than 12 weeks, ability to communicate in Hokkien or Mandarin, and willingness to participate. Exclusion criteria were serious mental problems, including dementia, inability to communicate cogently, and inability to walk to the community care center.

### 2.3. Measure

The Older Adults’ Health Promotion Activity Questionnaire (see Appendix A) was developed based on a comprehensive systematic review, following interviews with 47 older adults who had been randomly selected and a panel of four experts (two academics in health promotion and two gerontologists). Additionally, 12 older adults, who were not invited to the interview, were recruited to review the questionnaire for readability and comprehension. The questionnaire comprises three parts. The first part included demographic data such as gender, age, marital status, educational level, income, self-reported health status, and chronic disease. The second part was designed to explore the frequency of older adults’ participation in community-based health promotion activities (1 item). The respondents were asked to indicate their frequency using a 5-point Likert scale (very rarely = 1, rarely = 2, occasionally = 3, frequently = 4, very frequently = 5). The third part was the key part based on a five-dimensional Pender’s Health Promotion Model. It included statements regarding perceived benefits (19 items), perceived barriers (20 items), self-efficacy (10 items), social support (14 items), and activity-related affect (9 items). All items were rated on a 5-point Likert scale ranging from 1 (completely disagree/not confident at all) to 5 (completely agree/very confident). Higher scores indicate better benefits, self-efficacy, social support, and positive affects. All negatively worded items in the questionnaire were reverse-coded.

### 2.4. Validity and Reliability

The validity and reliability of the questionnaire were assessed. The content validity and face validity of the questionnaire were verified using the content validity index (CVI), based on ratings of item relevance by a panel of four experts. No item was eliminated in the CVI assessment, and all items had a score above 0.88. Only four items were revised for appropriateness. The preliminary questionnaire was pilot tested with 42 older adults. Cronbach’s alpha coefficient was used to evaluate the stability and internal consistency of the instrument. Cronbach’s alpha coefficients were reported ranging from 0.72 to 0.94 for all subscales (perceived benefits: α = 0.94; perceived barriers: α = 0.91; self-efficacy: α = 0.84; social support: α = 0.72; activity-related affect: α = 0.94), indicating acceptable level of internal consistency for each domain.

### 2.5. Statistical Analyses

Statistical analyses were performed using IBM SPSS Statistics for Windows, Version 26 (SPSS Inc., Chicago, IL, USA). Descriptive statistics (percentages of frequencies, means, and standard deviations) were calculated. An independent-samples *t*-test was used for the comparison of two independent groups; the comparison of three or more groups was performed using analysis of variance (ANOVA) test statistics. Pearson’s correlation analysis was conducted between older adults’ participation in community-based health promotion activities and the Health Promotion Questionnaire with all subscales. A multiple linear regression analysis was used to examine predictors associated with older adults’ participation in community-based health promotion activities. P values less than 0.05 were considered statistically significant for all tests.

## 3. Results

The participants’ mean age was 72.7 years (range: 65–88), and 74.8% were females. Of all the elderly individuals, 64.7% were married, 59% had a lower level of education, 51.8% had lower income from their pension, and 59% reported their health status as not good. The majority of the elderly people participating in the study (76.3%) had chronic diseases. Mean scores of the Health Promotion Questionnaire with all subscales is shown in Table 1.

There was a significant correlation between the participants’ perceived benefits mean scores and their gender. It was found that perceived benefits mean scores were significantly higher in females in comparison to males (t = −1.351, *p* = 0.033). Perceived barriers mean scores of those who were divorced or single (never married) were significantly higher than those who were married (F = 2.168, *p* = 0.002). There was a significant correlation between the participants’ self-efficacy mean scores, their gender, and self-reported health status. It was found that self-efficacy mean scores were significantly higher in females in comparison to males and in those who had reported health status as good in comparison to those who had reported their health status as not good (t = −1.725, *p* = 0.047; t = −4.622, *p* = 0.004). Activity-related affect mean scores of those who had a lower education level were significantly higher than those who had a higher education level (t = 0.434, *p* = 0.021). No significant correlations were found between social support mean scores and participants’ characteristics (*p* > 0.05) (see Table 1).

The Health Promotion Questionnaire included all subscales and participation in community-based health promotion activities using Pearson’s correlation analysis. The results revealed that older adults’ participation in community-based health promotion activities was significantly correlated with perceived benefit (*r* = 0.22, *p* < 0.05) and self-efficacy (*r* = 0.17, *p* < 0.05) (see Table 2).

A multiple linear regression analysis was performed to examine predictors associated with participation in community-based health promotion activities among the older adults. Multicollinearity among the independent variables was examined using correlation coefficients and variance inflation factor (VIF). No extreme coefficient value > 0.8 was found between the independent variables, indicating a low risk of multicollinearity. All independent variables had VIF ≥ 10 and tolerance ≥ 0.1, indicating no presence of multicollinearity. All variables, including the Health Promotion Questionnaire with all subscales, as well as the demographic characteristics of the participants, were entered as independent variables to predict the dependent variable, namely, older adults’ participation in community-based health promotion activities. Age, perceived benefits, and self-efficacy were identified as significant predictors of older adults’ participation in community-based health promotion activities (β = 0.202, *p* < 0.05, β = 0.305, *p* < 0.05, β = 0.060, *p* < 0.05, respectively). Among all the significant predictors, perceived benefits had the highest standardized regression coefficient (β= 0.305; *p* < 0.05) indicating participants who perceived higher benefits were more likely to have a higher participation in health promotion activities. Further, age and self-efficacy demonstrated relative higher contributions toward participation in health promotion activities. The results suggested that significant variables in the Pender’s Health Promotion Model were important in predicting the factors related to older adults’ participation in community-based health promotion activities. The model was significant and explained 35.3% of the variance of older adults’ participation in community-based health promotion activities (F =2.293, df = 8, 129, *p* < 0.05) with an adjusted R^2^ of 0.249 (see Table 3).

## 4. Discussion

The aim of this study was to apply the Pender’s Health Promotion Model to identify the factors associated with older adults’ participation in community-based health promotion activities. This study found that perceived benefits were the strongest predictor (β = 0.305; *p* < 0.05), with participants being more likely to engage in health promotion activities if their perceived benefits are high. Studies with systematic reviews have shown that the important predictor was typically perceived benefits when assessing the adherence of participants attending community-based exercise programs [8,20]. A study has been conducted to determine the effect of a multi-strategy program based on the Pender’s Health Promotion Model, to prevent loneliness of elderly women by improving social relationships. The results showed that perceived benefits and barriers were significant variables related to reducing loneliness in older women [21]. Furthermore, another study [22] on older adults’ health beliefs regarding the motivation to exercise, perceived benefits and barriers were the most direct determinants of increasing a high continuous participation rate. Although perceived barriers were not a significant predictor in our study, there is a need to focus on increasing awareness of community-based health promotion activities benefits while reducing the identified barriers.

Self-efficacy is an important determinant for complex activities and long-term changes in health behaviors [23]. This study is consistent with previous studies that cited the importance of self-efficacy in health promotion activities in older adults and demonstrated that older adults who have more confidence, are more competent to manage their health, and are more likely to regularly engage in health promotion activities [3,7]. Studies also found that self-efficacy is an indicator for predicting important health outcomes such as healthy eating, oral health, and hypertension prevention in different populations [24,25]. The findings support the importance of self-efficacy for engaging in community-based activities and should be considered in interventions to increase the continuous participation rate [7].

The results include significant differences between participants’ characteristics and the Health Promotion Questionnaire with all subscales included gender, marital status, educational level, and self-reported health status in this study. It was found that perceived benefits and self-efficacy mean scores were significantly higher in females in comparison to males and this is consistent with the findings of Seoa [26]. Studies found that women have a higher level of health knowledge and are more active in seeking health-related information than men do, which is thought to account for higher health-seeking behaviors [27]. Thus, gender is an important role in promoting health behavior. Meanwhile, we have found self-efficacy mean scores were significantly higher in those who had self-reported their health status as good in comparison to those who had self-reported their health status as not good. Since older people who perceive good health status tend to have higher self-efficacy, they may be more capable of looking after themselves and be more active in leisure activities, housework, and functional activities [9].

Married participants perceived significantly lower barriers than those who were divorced or single (never married) in this study. Married people may have the support from their spouse, so they perceive a lower level of barriers in participation of health promotion activities. Zhuori [28] suggests that with the support of family members, friends, and the public, older adults may be encouraged to participate in activities by attending recreational exercises, which in turn facilitates them to return to the society. Activity-related affect mean scores of those who had a lower education level were significantly higher than those who had a higher education level in this study, which is opposite to the findings of a previous study [29] and this suggested that it might be important to assess the impact of the interaction between self-efficacy and affect to ensure an effective health promotion program in further studies.

This study has some limitations. Data collection was limited to a particular care center in Taoyuan City for sampling convenience, resulting in a small sample size of merely 139 participants, who were surveyed only once. The data were cross-sectional which precludes inferences related to factors that affect older adults’ long-term participation in community-based health promotion activities. Moreover, as data were self-reported, there is a risk of self-report bias including social desirability and introspective ability. Further studies using experimental designs are needed to test causality in the associations among the measured variables in this study.

The results of this study show that the constructs of Pender’s Health Promotion Model can be used as a framework for predicting and detecting significant factors related to older adults’ participation in community-based health promotion activities. By using this model as a framework, researchers can design more specific studies that are directed towards improving a healthy lifestyle and detecting the key components of health-related behaviors among different age groups.

## 5. Conclusions

To conclude, age, perceived benefits, and self-efficacy were identified as significant predictors of older adults’ participation in community-based health promotion activities. Older adults perceived health promotion activities as beneficial, which in turn encouraged them to participate in them, resulting in a high participation rate. With an increase in older adults’ self-efficacy, they are more likely to regularly engage in health promotion activities. The results of this study can serve as a reference when developing health promotion plans for older adults.

## Figures and Tables

**Table 1 ijerph-18-09985-t001:** Comparison of participants’ characteristics and the Health Promotion Questionnaire with all subscales (*N* = 139).

Variables			Perceived Benefits		Perceived Barriers		Self-Efficacy		Social Support		Activity-Related Affect	
	*n*	%	Mean	SD	Mean	SD	Mean	SD	Mean	SD	Mean	SD
Total			4.19	0.45	1.79	0.53	3.92	0.83	3.76	0.53	4.48	0.49
Gender												
Male	35	25.2	4.10	0.37	1.71	0.51	3.72	1.01	3.74	0.54	4.44	0.46
Female	104	74.8	4.22	0.48	1.81	0.53	3.99	0.74	3.76	0.53	4.49	0.49
Statistical Analysis *p* value			t = −1.351 *p* = 0.033 *		t = −0.983 *p* = 0.720		t = −1.725 *p* = 0.047 *		t = −0.230 *p* = 0.761		t = −0.494 *p* = 0.680	

Age						-						
65–74	93	66.9	4.14	0.44	1.83	0.54	3.86	0.85	3.75	0.50	4.44	0.50
>75	46	33.1	4.28	0.46	1.69	0.50	4.06	0.76	3.76	0.61	4.57	0.46
Statistical Analysis *p* value			t = −1.732 *p* = 0.310		t = 1.472 *p* = 0.932		t = −1.360 *p* = 0.186		t = −0.103 *p* = 0.112		t = −1.467 *p* = 0.148	

Marital status												
Married	90	64.7	4.17	0.44	1.74	0.50	3.94	0.87	3.78	0.51	4.48	0.46
Widowed	41	29.5	4.25	0.49	1.81	0.61	3.93	0.74	3.75	0.60	4.51	0.54
Divorced/Single	8	5.8	4.13	0.39	2.14	0.22	3.68	0.81	3.57	0.42	4.31	0.45
Statistical Analysis *p* value			F = 0.473 *p* = 0.624		F = 2.168 *p* = 0.02 * (3>1)		F = 0.386 *p* = 0.680		F = 0.546 *p* = 0.581		F = 0.618 *p* = 0.541	

Education status												
Less than high school	82	59	4.16	0.49	1.77	0.54	3.97	0.76	3.75	0.56	4.49	0.52
Higher education level	57	41	4.23	0.40	1.80	0.51	3.86	0.91	3.77	0.50	4.46	0.44
Statistical Analysis *p* value			t = −0.956 *p* = 0.393		*t* = −0.274 *p* = 0.674		t = 0.783 *p* = 0.319		t = −0.303 *p* = 0.281		t = 0.434 *p* = 0.021 *	

Monthly income												
Below USD 300	72	51.8	4.26	0.46	1.66	0.50	4.00	0.86	3.71	0.58	4.57	0.47
Over USD 300	67	48.2	4.11	0.43	1.92	0.52	3.84	0.78	3.81	0.47	4.38	0.49
Statistical Analysis *p* value			t = 1.951 *p* = 0.336		t = −3.017 *p* = 0.544		t = 1.089 *p* = 0.869		t = −1.021 *p* = 0.134		t = 2.253 *p* = 0.284	
Self-reported health status												
Not good	82	59	4.16	0.48	1.84	0.56	3.67	0.86	3.68	0.52	4.46	0.50
Good	57	41	4.23	0.41	1.71	0.47	4.29	0.61	3.86	0.55	4.51	0.48
Statistical Analysis *p* value			t = −0.856 *p* = 0.416		t = 1.462 *p* = 0.237		t = −4.622 *p* = 0.004 *		t = −1.962 *p* = 0.893		t = −0.628 *p* = 0.910	

Chronic disease												
No	33	23.7	4.19	0.51	1.67	0.42	4.11	0.75	3.75	0.56	4.52	0.53
Yes	106	76.3	4.19	0.44	1.82	0.55	3.87	0.84	3.76	0.53	4.47	0.47
Statistical Analysis *p* value			t = 0.041 *p* = 0.478		t = −1.415 *p* = 0.154		t = 1.484 *p* = 0.202		t = −0.096 *p* = 0.790		t = 0.480 *p* = 0.244	


USD: United States dollar; * *p* < 0.05.

**Table 2 ijerph-18-09985-t002:** Pearson’s correlation analysis between participation in community-based health promotion activities and the Health Promotion Questionnaire with all subscales.

	Perceived Benefits	Perceived Barriers	Self-Efficacy	Social Support	Activity-Related Affect
*r*	0.22	−0.04	0.17	0.02	0.12
*p*	0.011 *	0.640	0.036 *	0.811	0.155

* *p* < 0.05.

**Table 3 ijerph-18-09985-t003:** Multiple regression analysis model examining predictors associated with participation in community-based health promotion activities (*N* = 139).

Variables	B	Std. Error	Beta	t	*p*-Value
Age	0.696	0.298	0.202	2.337	0.021 *
Gender	0.065	0.325	0.018	0.201	0.841
Self-reported health status	0.261	0.300	0.079	0.870	0.386
Perceived Benefits	1.090	0.425	0.305	2.566	0.011 *
Perceived Barriers	−0.441	0.331	−0.144	−1.333	0.185
Self-efficacy	0.118	0.184	0.060	0.644	0.041 *
Social Support	−0.370	0.296	−0.122	−1.253	0.212
Activity-related Affect	0.070	0.369	0.021	0.189	0.850

R^2^ = 0.353 (35.3%), adjusted R^2^ = 0.249 (24.9%), * *p* < 0.05.

## Data Availability

Research data are not available because participant consent did not include sharing of data.

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
