# Peer review of "Applying the Pender’s Health Promotion Model to Identify the Factors Related to Older Adults’ Participation in Community-Based Health Promotion Activities"

_ijerph, 2021, doi:10.3390/ijerph18199985_

Round 1
Reviewer 1 Report
I would like to congratulate the authors for their work.
The authors have made the changes proposed in previous revisions. However, they have eliminated the chapter on conclusions (chapter that was available in previous versions).
This chapter should be included in the document as a synthesis.
I believe that, with this minor change, the article will be significantly improved.
Author Response
Response to Reviewer 1 Comments
Reviewer 1.
I would like to congratulate the authors for their work.
Point 1: The authors have made the changes proposed in previous revisions. However, they have eliminated the chapter on conclusions (chapter that was available in previous versions).
This chapter should be included in the document as a synthesis.
I believe that, with this minor change, the article will be significantly improved.
Response 1: Thank you so much for giving me another opportunity to revised this manuscript. Conclusions are added on page 7-8 in red color.

Reviewer 2 Report
This paper suggests a model for identifying factors related to older adults’ participation in health promotion community activities. The manuscript needs to improve in organizational terms. Some important sections should be included, e.g., related works. The work methodology also needs to be better explored by the authors. Here are some suggestions for improving the study.
1) Include a paragraph about the paper's organization in the Introduction. Highlight the study main contributions in novelty terms;
2) Authors used Pender’s model to conduct their study. Is this model developed in the 1980s not outdated? How do the authors justify its use in this research?
3) The authors used a multiple linear regression analysis was performed to examine predictors. Have other statistical methods been tested to justify this approach?
4) Some references used in the manuscript are old. Authors should replace them with recent studies of the current state of the art.
Author Response
Response to Reviewer 2 Comments
Reviewer 2
This paper suggests a model for identifying factors related to older adults’ participation in health promotion community activities. The manuscript needs to improve in organizational terms. Some important sections should be included, e.g., related works. The work methodology also needs to be better explored by the authors. Here are some suggestions for improving the study.
Point 1) Include a paragraph about the paper's organization in the Introduction. Highlight the study main contributions in novelty terms;
Response 1: Thank you for your comments. A paragraph has been added to highlight the main contributions in this study (2nd paragraph in introduction section).
Point 2) Authors used Pender’s model to conduct their study. Is this model developed in the 1980s not outdated? How do the authors justify its use in this research?
Response 2: Thank you for your comments. Pender's health promotion model identifies factors that influence health behavior. The reason for the use of this model is its comprehensiveness and application in recognizing the determinants of behavior. We use Pender’s health promotion model because this model explores, from a theoretical perspective, the factors and relationships that contribute to participation in community-based health promotion activities and enhanced health and quality of life among older adults.
Although this model was developed in 1987, recent studies have found the strongest predictors in nutritional and self-care behaviors [1,2]. The justification of use Pender's health promotion model was added on 3rd paragraph in introduction section.
References
- Haghi, R.; Ashouri, A.; Karimy, M.; Rouhani-Tonekaboni, N.; Kasmaei, P.; Pakdaman, F.; Zareban, I. The role of correlated factors based on Pender health promotion model in brushing behavior in the 13–16 years old students of Guilan, Iran. Ital J Pediatr. 2021, 47, 1-10.
- Pouresmali, A.; Alizadehgoradel, J.; Molaei, B.M.V.; Fathi, D. Self-care behavior prevention of Covid-19 in the general population based on Pender health promotion model. Re Sq, 2020, 1-17.
Point 3) The authors used a multiple linear regression analysis was performed to examine predictors. Have other statistical methods been tested to justify this approach?
Response 3: Thank you for your comments. Correlation coefficients and variance inflation factor (VIF) were used to detect multicollinearity in multiple linear regression analysis. No extreme coefficient value > 0.8 was found between the independent variables, indicating a low risk of multicollinearity. All independent variables had VIF > 10 and tolerance > 0.1, indicating no presence of multicollinearity. We have added this justification on 4st paragraph in results section.
Point 4) Some references used in the manuscript are old. Authors should replace them with recent studies of the current state of the art.
Response 4: Thank you for your comments. Some outdated references were deleted and recent studies were added in the manuscript.
Round 2
Reviewer 2 Report
The authors accepted all suggestions from this reviewer. I consider the paper ready for publication.
This manuscript is a resubmission of an earlier submission. The following is a list of the peer review reports and author responses from that submission.
Round 1
Reviewer 1 Report
This article represents important work to identify barriers and benefits of health promotion activities for older adults. The results shared have the potential to build knowledge and improve application of health promotion to improve outcomes.
Introduction:
Clear identification/age parameters for “aging population” needed early in background since this could vary by organization or perspective or even across countries/systems based on life expectancy and rates of disability. A statistic is quoted related to 65+ but this is not clearly defined as the population of interest.
Recommend “The adverse health outcomes are…” be edited to clearly identify meaning in this context…possibly “Adverse health outcomes such as XXX and XXX are considerably…” or “The increased healthcare needs of an aging population are”
Areas of background need to be clarified in describing prior findings vs. conclusions (ex. Pg. 2 “Given that continuously engaging…”, a careful reword would better fit findings to evidence base: “Engagement in health promotion activities lasting 12 weeks of longer is a critical component to lasting health effects as such activities have been reported to have significant positive effects on older adults’ physical, mental, and spiritual well-being.”)
Also, review introduction for areas to improve clarity of sentences-reduce/break up longer sentences to improve readability and focus each sentence’s purpose (ex. Last full paragraph on pg. 2 “Students have further indicated that participating….” Needs to be broken into 2 thoughts or made more concise)
Methods:
It is unclear why a completely new assessment tool was created vs. using existing measures of these items or adapting. Solid description of processes for development, reliability and validity are provided but understanding of why a completely new tool was needed would strengthen this section. Is it because other tools didn’t target the aging population in their development or another reason?
Results:
Clarification needed around discussion of results on perceived barriers in lines 7-10 are somewhat unclear. Clarify further what is meant by items that “indicate fewer barriers” and how that results in older adults “perceived fewer barriers.” The subsequent examples assist in clarity but initial description somewhat confusing.
Recommend providing literature supported strategies for addressing the promotion activity barrier around poor physical health and how to foster that social/family support important to attendance. As it stands, the results are provided but without a sense that there are solutions available or that have been previously identified. If none have been identified for either significant finding on barrier/support, this needs to be clearly stated and identified as a needs area of future research. It would short-change the impact of the article’s contribution to not draw the connection to solutions. Contribution does not need to be limited to needing longitudinal data-this would address one angle of needed information. The barriers identified can have solutions even in short term and should be pulled forward to increase the implementation and actionability of this work.
Typos:
pg 9 within paragraph, Pearson’s
Abstract-line 6---affect should be changed to effect in this use/context (affecting describes process of it occurring; effect is the noun describing the end result)
Author Response
Response to Reviewer 1 Comments
Reviewer 1
This article represents important work to identify barriers and benefits of health promotion activities for older adults. The results shared have the potential to build knowledge and improve application of health promotion to improve outcomes.
Response: Thank you so much for giving us an opportunity to revise this manuscript. We did learn a lot form the three reviewers. Also, we tried very hard to revise the following comments. All revisions are on red color. Thanks again for the valuable comments.
Introduction:
Point 1: Clear identification/age parameters for “aging population” needed early in background since this could vary by organization or perspective or even across countries/systems based on life expectancy and rates of disability. A statistic is quoted related to 65+ but this is not clearly defined as the population of interest.
Response 1: Thank you for your comments. We have revised the sentence to make it clearer (1st paragraph in introduction section).
Point 2: Recommend “The adverse health outcomes are…” be edited to clearly identify meaning in this context…possibly “Adverse health outcomes such as XXX and XXX are considerably…” or “The increased healthcare needs of an aging population are”
Response 2: Thank you for your comments. We have revised the sentence to make it clearer (1st paragraph in introduction section).
Point 3: Areas of background need to be clarified in describing prior findings vs. conclusions (ex. Pg. 2 “Given that continuously engaging…”, a careful reword would better fit findings to evidence base: “Engagement in health promotion activities lasting 12 weeks of longer is a critical component to lasting health effects as such activities have been reported to have significant positive effects on older adults’ physical, mental, and spiritual well-being.”)
Response 3: Thank you for your comments. We have revised the sentence to make it clearer (3rd paragraph in introduction section).
Point 4: Also, review introduction for areas to improve clarity of sentences-reduce/break up longer sentences to improve readability and focus each sentence’s purpose (ex. Last full paragraph on pg. 2 “Students have further indicated that participating….” Needs to be broken into 2 thoughts or made more concise)
Response 4: Thank you for your comments. We have revised the sentences to make them clearer (2nd paragraph in introduction section).
Methods:
Point 5: It is unclear why a completely new assessment tool was created vs. using existing measures of these items or adapting. Solid description of processes for development, reliability and validity are provided but understanding of why a completely new tool was needed would strengthen this section. Is it because other tools didn’t target the aging population in their development or another reason?
Response 5: Thank you for your comments. There is no health promotion activity questionnaire designed for older adults. Therefore the research team decided to develop a new tool for aging population.
Results:
Point 6: Clarification needed around discussion of results on perceived barriers in lines 7-10 are somewhat unclear. Clarify further what is meant by items that “indicate fewer barriers” and how that results in older adults “perceived fewer barriers.” The subsequent examples assist in clarity but initial description somewhat confusing.
Response 6: Thank you for your comments. We have revised the sentence as ‘The mean score for the perceived barriers to health promotion activities domain was 1.79 (SD = 0.53), indicating that older adults had low perceived barriers and were committed to participating in health promotion activities.’ to make it clearer (1st paragraph in result section).
Point 7: Recommend providing literature supported strategies for addressing the promotion activity barrier around poor physical health and how to foster that social/family support important to attendance. As it stands, the results are provided but without a sense that there are solutions available or that have been previously identified. If none have been identified for either significant finding on barrier/support, this needs to be clearly stated and identified as a needs area of future research. It would short-change the impact of the article’s contribution to not draw the connection to solutions. Contribution does not need to be limited to needing longitudinal data-this would address one angle of needed information. The barriers identified can have solutions even in short term and should be pulled forward to increase the implementation and action ability of this work.
Response 7: Thank you for your comments. We have revised to make them clearer (4th paragraph in discussion section). “ However, no long-term, continuous data are available to determine whether participation would be affected by older adults’ increasing age or other factors. In the future, long-term changes in older adults’ participation in community-based health promotion activities should be explored through longitudinal studies to better understand the factors affecting participation.” have been deleted.
Point 8: Typos:
pg 9 within paragraph, Pearson’s
Response 8: Thank you for the correction. It is corrected on page 9.
Point 9: Abstract-line 6---affect should be changed to effect in this use/context (affecting describes process of it occurring; effect is the noun describing the end result)
Response 9: Thank you for the correction. According to reviewer 2 comments, the abstract has been revised in red color.

Reviewer 2 Report
I would like to congratulate the authors for their interest in researching in this field, however, the work presented presents some deficiencies.
- The keywords are insufficient and do not accurately reflect the main identifiable contents of the article. This section should be revised and adapted to the needs of the article.
Keywords should comply with:
-Enable the reader to identify the topic easily.
-Allow a precise indexing of the material.
- The abstract does not provide information for each part of the work, so it would not comply with the fundamental information of a research summary. More specifically, you should briefly explain the results should describe the results in a summarized yet specific manner.
- Section 2.1 should explicitly indicate the characteristics of the center in which the experiment was carried out (type, size, etc.). The authors should indicate whether the type of center affects the data obtained in the experiment and, consequently, whether the experiment can be extrapolated to other centers with similar characteristics. However, I believe that they should carry out an analysis of m comparisons. However, the data that characterize the participants, such as their age, sex, etc., are not indicated.
- In section 2.4, the authors indicate that they will use Cronbach's alpha coefficient to evaluate the stability and internal consistency of the instrument. They also indicate that the values obtained were, for all categories, higher than 0.84. However, a detailed analysis of the data, allow us to observe that in the case of the category "social support", the coefficient α = 0.72; This data is relatively low. They should justify the suitability of the case studied by also taking into account this result.
- In section 2.5, the authors indicate that they will perform a statistical analysis of the data using Pearson's correlation coefficient. This index is a measure of linear dependence between two quantitative random variables. However, I consider that they should perform a multiple comparisons analysis for the different categories studied in order to determine that there is no correlation between the different families of data analyzed that could affect the interpretation of the results. If the authors consider that this test is unnecessary, please justify their decision.
- Conclusions chapter is correct, although they should be based on the reflections in the Discussion section. For this reason, after the reform of this section, the Conclusions section should be revised.
I hope that these changes will help to improve your article and make it a document of great scientific interest.
Author Response
Response to Reviewer 2 Comments
I would like to congratulate the authors for their interest in researching in this field, however, the work presented presents some deficiencies.
Response: Thank you so much for giving us an opportunity to revise this manuscript. We did learn a lot form the three reviewers. Also, we tried very hard to revise the following comments. All revisions are on red color. Thanks again for the valuable comments.
Point 1: - The keywords are insufficient and do not accurately reflect the main identifiable contents of the article. This section should be revised and adapted to the needs of the article.
Keywords should comply with:
-Enable the reader to identify the topic easily.
-Allow a precise indexing of the material.
Response 1: Thank you for your comments. Keywords have revised as Pender’s health promotion model, older adults, community-based health promotion activities
Point 2:- The abstract does not provide information for each part of the work, so it would not comply with the fundamental information of a research summary. More specifically, you should briefly explain the results should describe the results in a summarized yet specific manner.
Response 2: Thank you for your comments. The abstract has revised to make it clearer on page 1.
Point 3: - Section 2.1 should explicitly indicate the characteristics of the center in which the experiment was carried out (type, size, etc.). The authors should indicate whether the type of center affects the data obtained in the experiment and, consequently, whether the experiment can be extrapolated to other centers with similar characteristics. However, I believe that they should carry out an analysis of m comparisons. However, the data that characterize the participants, such as their age, sex, etc., are not indicated.
Response 3: Thank you for your comments.
- The characteristics of the center has been added in Section 2.1
- The generalizability of this study is addressed in 5th paragraph in discussion section (p. 11).
- Participants’ characteristics is illustrated in Table 1 including gender, age range, marital status, self-reported health status, education status, chronic disease and frequency of participating in health promotion activities per week.
Point 4: - In section 2.4, the authors indicate that they will use Cronbach's alpha coefficient to evaluate the stability and internal consistency of the instrument. They also indicate that the values obtained were, for all categories, higher than 0.84. However, a detailed analysis of the data, allow us to observe that in the case of the category "social support", the coefficient α = 0.72; This data is relatively low. They should justify the suitability of the case studied by also taking into account this result.
Response 4: Thank you for your comments. Cronbach’s alpha coefficients in social support was
0.72 and this was revised in 2.4 Validity and Reliability section. Cronbach’s alpha coefficients in
social support was lower when comparing with other subscales. However, reference about the
acceptable values of alph was ranging from 0.70 to 0.95 [1,2]. Therefore, cronbach’s alpha
coefficients in social support is greater than 0.72 which is acceptable. Values of alpha is shown in Table 1 (in attached file)
References
- Tavakol, M.; Dennick, R. Making Sense of Cronbach’s Alpha. Int J Medical Educ. 2011, 2, 53-55.
- Salkind, N. Encyclopedia of Measurement and Statistics1st Edition. SAGE: USA
2015; pp.126-130.
Point 5: - In section 2.5, the authors indicate that they will perform a statistical analysis of the data using Pearson's correlation coefficient. This index is a measure of linear dependence between two quantitative random variables. However, I consider that they should perform a multiple comparisons analysis for the different categories studied in order to determine that there is no correlation between the different families of data analyzed that could affect the interpretation of the results. If the authors consider that this test is unnecessary, please justify their decision.
Response 5: Thank you for this suggestion. In our study, the sample size is 139 when a multiple comparison analysis is performed, a group has different sample sizes. This will give a result which may not be sufficiently powered to detect a difference between the groups and the study may turn out to be falsely negative leading to a type II error [1,2]. We do hope the justification is accepted. Thanks again for this good suggestion, more participants will be invited and multiple comparisons tests will be adopted in the future study.
References
- Lee, S.; Kee, D. K. What is the proper way to apply the multiple comparison test?
Korean J Anesthesiol. 2018, 71, 353–360.
- Barun Kumar Nayak, B. K. Understanding the relevance of sample size calculation Indian J Ophthalmol. 2010, 8, 469–470.
Point 6:- Conclusions chapter is correct, although they should be based on the reflections in the Discussion section. For this reason, after the reform of this section, the Conclusions section should be revised.
I hope that these changes will help to improve your article and make it a document of great scientific interest.
Response 6: Thank you for your comments. According to the response 5, conclusions chapter is not revised. Many thanks for the valuable comments.
Point 4: Also, review introduction for areas to improve clarity of sentences-reduce/break up longer sentences to improve readability and focus each sentence’s purpose (ex. Last full paragraph on pg. 2 “Students have further indicated that participating….” Needs to be broken into 2 thoughts or made more concise)
Response 4: Thank you for your comments. We have revised the sentences to make them clearer (2nd paragraph in introduction section).
Methods:
Point 5: It is unclear why a completely new assessment tool was created vs. using existing measures of these items or adapting. Solid description of processes for development, reliability and validity are provided but understanding of why a completely new tool was needed would strengthen this section. Is it because other tools didn’t target the aging population in their development or another reason?
Response 5: Thank you for your comments. There is no health promotion activity questionnaire designed for older adults. Therefore the research team decided to develop a new tool for aging population.
Results:
Point 6: Clarification needed around discussion of results on perceived barriers in lines 7-10 are somewhat unclear. Clarify further what is meant by items that “indicate fewer barriers” and how that results in older adults “perceived fewer barriers.” The subsequent examples assist in clarity but initial description somewhat confusing.
Response 6: Thank you for your comments. We have revised the sentence as ‘The mean score for the perceived barriers to health promotion activities domain was 1.79 (SD = 0.53), indicating that older adults had low perceived barriers and were committed to participating in health promotion activities.’ to make it clearer (1st paragraph in result section).
Point 7: Recommend providing literature supported strategies for addressing the promotion activity barrier around poor physical health and how to foster that social/family support important to attendance. As it stands, the results are provided but without a sense that there are solutions available or that have been previously identified. If none have been identified for either significant finding on barrier/support, this needs to be clearly stated and identified as a needs area of future research. It would short-change the impact of the article’s contribution to not draw the connection to solutions. Contribution does not need to be limited to needing longitudinal data-this would address one angle of needed information. The barriers identified can have solutions even in short term and should be pulled forward to increase the implementation and action ability of this work.
Response 7: Thank you for your comments. We have revised to make them clearer (4th paragraph in discussion section). “ However, no long-term, continuous data are available to determine whether participation would be affected by older adults’ increasing age or other factors. In the future, long-term changes in older adults’ participation in community-based health promotion activities should be explored through longitudinal studies to better understand the factors affecting participation.” have been deleted.
Point 8: Typos:
pg 9 within paragraph, Pearson’s
Response 8: Thank you for the correction. It is corrected on page 9.
Point 9: Abstract-line 6---affect should be changed to effect in this use/context (affecting describes process of it occurring; effect is the noun describing the end result)
Response 9: Thank you for the correction. According to reviewer 2 comments, the abstract has been revised in red color.

Reviewer 3 Report
The design of the study is not scientifically sound: you present cross-sectional statististics as a kind of post-test after a 12 weeks health promotion intervention, but did should at least be a pre- and post-test design and include a control group. The very descriptive Results focus on long tables of scores on item levels and one (post-test?) correlation matrix. In this way, the paper does not add much to existing literature and better designed intervention studies.
Author Response
Response to Reviewer 3 Comments
Point 1: The design of the study is not scientifically sound: you present cross-sectional statististics as a kind of post-test after a 12 weeks health promotion intervention, but did should at least be a pre- and post-test design and include a control group. The very descriptive Results focus on long tables of scores on item levels and one (post-test?) correlation matrix. In this way, the paper does not add much to existing literature and better designed intervention studies.
Thank you so much for giving us an opportunity to revise this manuscript. We did learn a lot form the three reviewers. Also, we tried very hard to revise the following comments. Thanks again for the valuable comments.
Response 1: Thank you for your comments. Our study is a descriptive study, cross-sectional design. This is not a pre-post-test study design. We would like to apply the Pender’s Health Promotion Model to identify the factors related to older adults’ participation in community-based health promotion activities. The participants participate in health promotion activities once a week and continuous participation for more than 12 weeks which is one of the inclusion criteria. Therefore descriptive results are illustrated in tables and the literature focus on elements of Pender’s Health Promotion Model.

Round 2
Reviewer 2 Report
I believe that the authors have addressed the requirements and implemented the corrections correctly.
Attending to punt 4, in section 2.4, the authors indicate that they will use Cronbach's alpha coefficient to evaluate the stability and internal consistency of the instrument. They also indicate that the values obtained were, for all categories, higher than 0.84. However, a detailed analysis of the data, allow us to observe that in the case of the category "social support", the coefficient α = 0.72;
The authors should correct this statement and indicate that all values he coefficient α are higher than 0.72 and not 0.84.
I trust that these corrections will help to achieve a document of greater scientific interest.
Best regards.
Reviewer 3 Report
Notwithstanding the response of the authors, I stick to my fundamental issue about the inadequate cross-sectional design in one single post test. The authors not even discuss this in their last section.